# A Preliminary Scoping Review of Trauma Recovery Pathways among Refugees in the United States

Crispin Rakibu Mbamba [1,*], Jennifer Litela Asare [2] and Clinton Gyimah [2]

1   School of Social Welfare, University at Albany, SUNY, Albany, NY 12222, USA
2   Department of Sociology and Social Work, Kwame Nkrumah University of Science and Technology, Kumasi 00233, Ghana
*   Correspondence: crispinrakim04@gmail.com

**Abstract:** When people move across borders to seek asylum because of violence, conflicts, persecution, or human rights violations, they experience a complex mix of psychological and traumatic downfalls. Often, refugees and asylum seekers' trauma is compounded by the behaviours of individuals, communities, and the systemic climate of host countries. The United States is host to refugees and asylees from several countries. Evidence shows that several asylum seekers are held up in deplorable conditions in immigration detention centres where they are battling acute trauma. Therefore, consequent to this, coupled with the varying trauma that refugees face, this preliminary scoping review explores the scope and context of available peer-reviewed scholarship on trauma recovery pathways among refugees in the United States to identify gaps for further research. Following the PRISMA-compliant scoping review guidelines, we identified and curated data on the scope and context of peer-reviewed literature on trauma recovery approaches among refugees in the United States. This study identified the following as trauma recovery pathways among refugees: (1) macro-level structural intervention—preventing re-traumatization; (2) culturally sensitive therapeutic intervention; and (3) diagnosis and therapy. This study concludes that little research on the recovery pathways among refugees exists in the United States, hence the need for scholarship in this area.

**Keywords:** trauma recovery; trauma recovery pathways; refugees; trauma; United States

## 1. Introduction

Due to conflict, violence, and persecution, there are currently more than 65 million individuals who have been rendered refugees globally. Every year, the US government decides how many refugees are allowed to enter the nation; these numbers can range from less than 30,000 to over 200,000 annually [1]. Refugees, asylum seekers, and other groups who have been forcibly displaced frequently describe exposure to numerous potentially traumatic events in both their home and host countries as well as when they were displaced [2]. This group's general health results are adversely affected by these encounters, which are frequently lengthy, repetitive, and interpersonal in nature. As a result, it has constantly been noted that migrants frequently report trauma-related issues, especially depression, anxiety, and post-traumatic stress disorder (PTSD) [3]. Refugees typically deal with several difficulties every day. These numerous challenges, which include those linked to a lack of resources, family division, social isolation, acculturation, prejudice, socioeconomic considerations, immigration, and refugee regulations are mostly tied to the post-migration environment [4]. Beyond the consequences of traumatic events, it has been demonstrated that these pressures associated with displacement have a significant influence on refugees' health [5,6]. Considering the widespread nature of these complications, this study curates and synthesizes evidence from peer-reviewed articles to ascertain the scope and context of trauma recovery pathways among refugees. For the purposes

of this study, we adopt the 1951 Refugee Convention's definition of a refugee that states that a refugee is someone who is unwilling or unable to return to their country of origin due to a well-founded fear of being persecuted for reasons of race, religion, nationality, membership of a particular social group, or political opinion (OHCHR, 1951). In addition, in establishing a better comprehension of the study, trauma recovery pathways deal with guiding principles of safety, restoration and empowerment that ultimately focus on the ability to live in the present without being overwhelmed by the thoughts, experiences, and feelings of the past, especially in their home country.

The complexity of post-migration-related concerns among traumatized refugees highlights the growing necessity for contextual information on recovery techniques and clinical interventions among these individuals that are essential to their survival [7]. This will primarily emphasize both mental and physical health, as well as enhance general health behaviour, daily life functioning, and psycho-social adjustment [8,9]. It is equally relevant to establish that the fundamental requirements of refugees, such as shelter, regular meals, protection from harm as well as the ability to work and earn income are crucial and inevitable in tackling the general wellbeing of refugees in trauma therapy and care [10]. These needs form a pivotal role in enabling refugees to stabilize and better appreciate and integrate into the new environment [11]. Typically, refugees must quickly integrate into their new society, especially in terms of their financial independence and language skills. However, the process of social integration necessitates highly functional standards in terms of cognitive and social skills as well as care systems available in host countries [12]. A recent study by Boettcher and Colleagues pointed out the relevance of physical activity as an unconsidered recovery mechanism for refugees and asylees, particularly in the US [13]. Physical activity (PA), specifically when practiced by asylees, is well-established as an efficient trauma reliever and has been linked to improved physical and mental wellbeing, decreased burden of pain and other somatic symptoms, increased cognitive and functional capacity, increased overall life satisfaction, and lowered risks of lifestyle diseases [14]. Interventions involving physical activity and exercise, such as aerobic conditioning, muscle strengthening, flexibility training, and movement therapy, are typically seen as crucial treatment facets in trauma-related illness among refugees in asylums [15]. Another analysis by von Haumeder and Colleagues also considered firm belief systems as a proven protective factor against post-traumatic disorders. It is important to recognize the impact of belief systems on how traumatized refugees respond to treatment mechanisms [16]. The use of narrative therapy has effectively been avouched as a recovery model for traumatised refugees, especially children. The demands of trauma survivors, particularly those impacted by war and torture, were accommodated by this methodology [17]. This method involves having patients discuss the most upsetting aspects of the trauma, which causes them to revisit the associated ideas and feelings [11]. The therapist duly probes the patient's account of the traumatic incident and, ultimately, reconstructs it while eliciting behavioural, physiological, cognitive, and emotional responses. Because of this, the capacity to create constructive narratives about traumatic events is correlated with a successful recovery process [18,19].

With respect to these mechanisms and other models of trauma recovery, this study seeks to explore and analyse the context and scope, with the intent of identifying and addressing gaps in the literature. In the subsequent paragraphs, the PRISMA-compliant scoping review guidelines will be duly followed to present empirical evidence and critiques on peer-reviewed literature on trauma recovery pathways. This is important in presenting a comprehensible structure, which will ultimately contribute to the optimum wellbeing and functioning of refugees, as they form an integral part of the global nexus and economic buoyancy. This review is guided by this question:

What is the scope and context of peer-reviewed literature on trauma recovery approaches among refugees in the United States?

## 2. Review Method

Sargeant and O'Connor [20] opined that scoping reviews are usually descriptive in nature. They stated that the essence of a scoping review is to map the literature in terms of geography and context around a topic of interest. Like a systematic review, a scoping review starts by extensively searching the literature. Scoping reviews are relevant for identifying gaps about a subject matter in the literature which creates attention, thus, informing research to bridge that gap.

The principal objective of the current scoping review is to map out the volume of peer-reviewed research (both original and review) with strict eligibility criteria on including articles that only report on trauma recovery pathways in the United States. This means, if a particular study reports on trauma recovery pathways in any country other than the United States (e.g., Australia), that study is excluded irrespective of the relevance of its content.

The expectation is that the studies that will be included address the research objective specific to the United States, and enunciate suggestions rooted in evidence for future research in the United States. In this study, we follow Arksey and O'Malley's six-step methodological framework for scoping reviews [21].

First, we searched academic databases including Google, Google Scholar, PubMed, Web of Science, and specific reputable journal sites using their search engines for articles that were published from 2000 to 2021. Journals such as Trauma Care and Child Abuse and Neglect were searched to identify studies that would be used for this study. The keyword and phrase combinations that were used to search, included "refugees AND trauma recovery", "asylum seekers and trauma", "Trauma and recovery", "detention centres and trauma recovery". Words such as "trauma", "Traumatic", "stress", and "anxiety" were matched with the names of all the states in the United Sates such as "New York", "Virginia", "Michigan", "Wisconsin" until all the 52 states were exhausted. In stage two of the search, keywords and phrases appeared in the titles which was the first point of screening. Abstracts of titles that looked appropriate were read. In stage three, only those articles that specifically reported on trauma recovery pathways among refugees and asylum seekers in the United States were maintained. In the fourth stage, all nine included peer-reviewed articles were fully read, charted, and collated as seen in Table 1. For stages five and six, we engaged in summarising and reporting the results to come up with this article. The figure below shows a summary of the article filter process leading to the included studies (Figure 1).

**Table 1.** Summary of included studies.

| Resource | Type of Article | Specific Methods | Age Group and Category | Location of Studies | Main Findings/Recommendations |
|---|---|---|---|---|---|
| [22] | Review | | | USA and Canada | • PTSD, acculturation, and minority stress provide therapists with a comprehensive and sufficient framework strategy for working with SGMAS. |
| [23] | Review | Thematic review | Youth and Families | Chicago | • The KCCTP quickly shifted to program accessibility, active outreach, costly case management, and flexible delivery of teletherapy and online psychological support.<br>• The importance of addressing structural barriers and basic needs was critical to family engagement and the therapeutic process. |
| [24] | Quantitative | 1 Multivariable linear regressions 2 Moderated regressions | 20–70 years old | USA | • Lower stress is associated with greater exposure to pre-migration trauma and changing to a more secure visa status.<br>• Having more access to social services and not reporting chronic pain are associated with lower PTSD. Stable housing and employment significantly moderate the relationship between lower chronic pain and lower PTSD. |
| [25] | | Conventional content analysis; Retrospective content analysis | | USA | • The most beneficial topic was trauma-informed care for refugee resettlement, and community partnership building was the most requested area for future training.<br>• Culturally responsive and trauma-informed approaches can help to bridge the gap between mental health care and resettlement services, as well as promote knowledge and specialist knowledge exchange in order to foster collaborative project care and community engagement. |
| [26] | Review | Narrative Review | | USA and other Western industrialized countries | • The article presents an overview of cultural influences in the treatment of refugee populations and introduces the multi-cultural orientation framework as a method to improve refugee mental health services and aid in the delivery of culturally appropriate care.<br>• It was resolved that it is critical for therapists to seek cultural prospects with humility and comfort in order to better serve the refugee population.<br>• It advances scholarly debate on the application of the multicultural orientation framework of psychotherapy to vulnerable populations such as refugees. |

**Table 1.** *Cont.*

| Resource | Type of Article | Specific Methods | Age Group and Category | Location of Studies | Main Findings/Recommendations |
|---|---|---|---|---|---|
| [27] | Quantitative | Cross-sectional study | Pregnant women who are 18 years and above | Atlanta, GA | • Pregnant women in high-risk communities require trauma and PTSD screening, increased awareness of the role trauma may play a role in perinatal depression, and referral to trauma-informed care. |
| [28] | Quantitative | Cross-sectional study | Adults | Massachusetts | • The study discovered that greater use of problem- or emotion-focused engagement coping strategies was associated with lower stress among refugees, implying that problem- or emotion- focused stress management interventions have the potential to reduce stress among refugees. |
| [29] | Review | Narrative Review | | Virginia | • The paper provided a framework for aligning stratified intervention and addressing refugee communities' multi-layer mental health needs.<br>• A multitier mental health psychosocial support model developed by an inter-agency standing committee was used in the study to provide a holistic framework for a care system.<br>• The study proposed a two-pronged approach to trauma-informed and culturally informed care that would be integrated into each tier of the MHPSS program. |
| [30] | Qualitative | Interviews | Youth 16–21 years | | • Narratives used/talked about anger, cognitive processes, discrepancy, tentativeness, perpetual processes, ingestion, relativity, time, work, and home<br>• The findings have implications for the efficacy of treating trauma symptoms with non-native language discourse. |

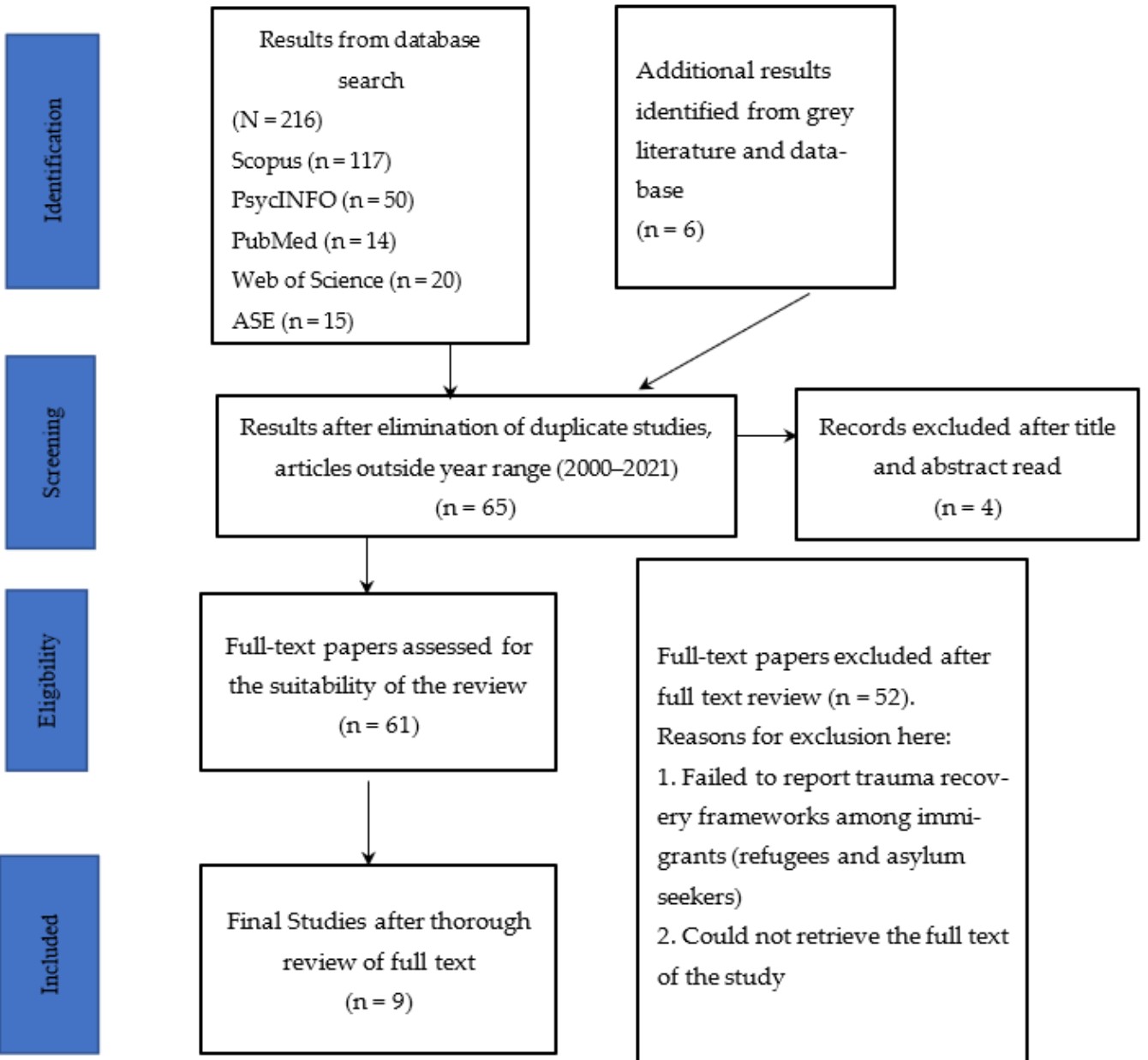

**Figure 1.** Summary of the article filter process leading to the included studies.

*Methodological Description of Included Studies*

Nine articles were included in this study. For the first article [22], the authors used a narrative review approach to present a framework for clinical practice with sexual and gender minority asylum seeker (SGMAS). The second article [23] uses a scientific commentary approach to review evidence on the Kovler Center Child Trauma Program (KCCTP) serving immigrant and refugee youth and families with trauma experience. Article three [24] uses archival clinical data to identify whether post-migration correlates with reductions in distress among torture survivors, after accounting for pre-migration trauma. The dataset consisted of 323 asylum seekers and refugees, consisting of 206 males and 117 females. To explore the associations between psychosocial treatment targets with changes in depression (PHQ-9) and PTSD (HTQ) scores, the study used multivariable linear regressions with the aid of SPSS software. The study also used tested moderated regressions in the SPSS software. Article four [25] developed and evaluated a program called Cross-Cultural Trauma-Informed Care (CC-TIC) training using free listing and semi-structured retrospective pre- and post-training evaluation in two unnamed states in the

United States. Article five [26] adopts a narrative review approach to curate evidence, and comes up with a culturally relevant framework to help treat psychological challenges among refugees in the United States and other Western industrialized countries that were not mentioned. For article six [27], the study consisted of 633 pregnant women in Atlanta, Georgia, and all participants had to be at least 18 years old. Analysis was conducted using descriptive statistics. In addition, analysis of variance (ANOVA) was conducted and chi-square tests were used to measure the differences in rates of trauma, PTSD, depression, and treatment engagement. Article seven [28] uses a sample of 8149, including refugees, individuals with special immigrant visas, and parolees/entrants in Massachusetts. Analysis was conducted using logistic regression. Article eight [29] uses a narrative review approach to provide a framework to align stratified interventions and address multi-layered mental health needs among refugees. Article nine [30] uses data from a large dataset that involved 10 participants aged 16 to 21, and the data were collected qualitatively using interviews. The 10 participants came from Pakistan, Turkey, Syria, Jordan, China, and The Republic of Congo. The participants were made up of 50% males and 50% females.

## 3. Results

### 3.1. Summary of Findings and Discussion

All studies that met eligibility totalled nine (n = 9) peer-reviewed journal articles. Included studies consisted of five (5) primary studies, which were either qualitative or quantitative papers, and four (4) review papers. All the papers reported on trauma recovery pathways among refugees in the United States of America. Specifically, evidence emerged from Chicago, Massachusetts and Virginia. All the other papers presented evidence from synthesising evidence across the United States without mentioning which states. Two (2) of the studies were joint studies involving the United States and Canada which researchers found relevant upon deliberation and, therefore, added to the review. Synthesised evidence for this review highlights three main areas where research on trauma recovery pathways in the United States has been focused on among refugees and asylum seekers: (1) Macro-level structural intervention—preventing re-traumatization. (2) Culturally sensitive therapeutic intervention. (3) Diagnosis and therapy.

### 3.2. Macro-Level Structural Intervention—Preventing Re-Traumatization

According to Alessi and Kahn [22], when individuals enter the United States, the chances that they have experiences of pre-migration trauma, especially refugees and those seeking asylum, are high. Therefore, some included studies proposed macro-level interventions such as focusing on the processes involved in managing asylum claims [22,24]. This will prevent them from experiencing the stress that comes with emigration processes, which will compound their already existing trauma. Very succinctly, Kashyap et al., refugees who are plagued with pre-migration trauma but manage to successfully change to a stable visa, have reduced stress rates [24]. In addition, community level interventions that prioritise putting asylum seekers and refugees in good transition centres instead of places such as detention camps with deplorable conditions [22–25] also reduce stress rates. Some of the studies specifically stressed the difficulty in acculturation (which comes with its own stress) and why it is important to strive to reduce the degree of challenges faced by refugees [22,23]. According to Endale and his colleagues for example [23], for a family or individual to be in a position to receive therapeutic services for mental health complications such as anxiety and stress, having a place to sleep and basic needs met is a necessity. They argue that trying to help people who are traumatised through counselling without first offering them a good environment to stay might not yield positive results.

### 3.3. Culturally Sensitive Therapeutic Intervention

Im and Swan focused on the need to make trauma-informed care specific to refugee resettlement, instead of providing services such as those given to locals [25]. Their study suggested that culturally responsive and trauma-informed approaches can help bridge the

gap between mental health care and resettlement services as well as promote the exchange of knowledge and expertise. According to Adams and Kivlighan [26], cultural consideration in the treatment of the refugee population is important, and a multi-cultural orientation framework is needed to augment mental health services for refugee populations in a way that is culturally appropriate. Adams and colleague again identified that even in therapy involving refugees in the United States, therapists need to seek cultural opportunities with humility to best serve the needs of the refugee population [26]. The essence of these revelations is captured in several other studies in many parts of Europe—studies involving refugees from Africa who enter Europe through the Mediterranean coast [31]. In some of these studies for example, despite the urgent need for help because of the daunting experiences of immigrants, some refugees still find it difficult to openly speak about their needs when asked at refugee centres [31]. Bokore's article on Suffering in silence revealed that a possible resulting factor is the culture of silence as an exhibition of respect for people of authority in some African communities [32]. Hence, there is the need to consider such prevailing cultural factors even in offering therapeutic services to refugees.

*3.4. Diagnosis and Therapy*

Some of the studies revealed that practitioners serving refugees and asylum seekers cannot always assume that because they are going through a crisis, they have any particular disorder or traumatic condition. Instead, there is the need to first, as therapists and counsellors, conduct appropriate diagnosis to identify the exact condition of individuals before any kind of treatment is given [27,28]. Oftentimes, wrong interventions are offered to immigrants [27]. For example, when individuals enter a country seeking asylum and struggling with immigration complexities and basic needs strategies (that is; food, shelter, safety, family reunification, transportation, employment, and migrant/refugee legal documentation) may precede and are of more significant urgency than trauma treatment and therapy, but chances are, there is usually an overemphasis on immediate counselling to the neglect of the conditions that compound and create new traumatic situations, such as the absence of survival needs.

Congruent with this notion, apart from the fact that therapy and supportive care must be trauma-informed, they must be rooted in evidence-based models of care [29]. Such identified models have proven to be effective with survivors of other types of traumas such as sexual assault, kidnapping, and physical or psychological abuse.

**4. Future Direction**

In the last decade, several people have been displaced and more refugees have entered the United States. Therefore, the limited research on trauma recovery across all states, calls for an increase in studies to properly understand the needs of refugees and asylum seekers, in order for service providers to have a yardstick to offer the best services. In addition, utilising ethnographic studies among refugee communities to effectively understand the situations of the group, when it comes to trauma recovery pathways will be essential, to enrich the scholarship in trauma care among refugees and improve services to this population. Finally, more discourse in the form of conferences relating to trauma recovery pathways, especially considering the systemic component will contribute to accumulating support and a bigger voice to advocate for the needs of refugees and traumatised asylees.

**5. Conclusions**

Some efforts have been made to conduct research on synthesising trauma recovery pathways among refugees and asylum seekers in the United States, and research efforts have been made in West Africa, especially in the area of professional knowledge; however, there are still gaps in scientific knowledge considering the scope of the available literature and the national geography. There is, therefore, the need to address these gaps and conduct further research pertaining trauma recovery among refugees (e.g., ethnographic studies of refugees across lifespan in the United States) with the help of critical methodologies.

**Author Contributions:** Conceptualization, C.R.M. and C.G.; methodology, C.R.M., C.G. and J.L.A.; validation, C.R.M., C.G. and J.L.A.; formal analysis, C.R.M., C.G. and J.L.A.; resources, C.R.M., C.G. and J.L.A.; data curation, C.R.M.; writing—original draft preparation, C.R.M., C.G. and J.L.A.; writing—review and editing, C.R.M., C.G. and J.L.A.; supervision, C.R.M.; project administration, C.R.M. All authors have read and agreed to the published version of the manuscript.

**Funding:** This research received no external funding.

**Institutional Review Board Statement:** Not applicable.

**Informed Consent Statement:** Not applicable.

**Data Availability Statement:** Not applicable.

**Conflicts of Interest:** The authors declare no conflict of interest.

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
