# Peer review of "A Preliminary Scoping Review of Trauma Recovery Pathways among Refugees in the United States"

_traumacare, doi:10.3390/traumacare2040048_

Round 1

Reviewer 1 Report

I was very excited to see the topic for this article and I would like to command the authors for undertaking this research and literature review, as it is important to assess trauma care responses for refugees and immigrants. However, this article did not go as deep as it should have done, even as a literature review article.  There were only 9 academic articles that made it through the elimination, some of which was avoidable (52 articles out of 61 articles eliminated because full-text was not available or because they did not refer to "trauma recovery frameworks?") and then those 9 articles and their findings were not discussed well or analyzed, but simply grouped under very general categories. Reading this article, or rather literature review, I did not learn anything new about the articles in question and was left with the feeling that my time would have been better spent reading the original articles.  This piece almost reads like an exercise in how to write a short literature review, without going into deeply the actual topic of trauma care for immigrants and refugees to pull important findings. I think part of the problem might be the broadness of the topic and the extra-focus on the geographical setting--if the authors were less worried about which state the article refers to and spent more effort on focusing the trauma experiences among one set of refugees or asylum seekers or immigrants (whose experiences of trauma might be varied depending on their displacement experiences), then we would have a stronger article.  With my regrets, I would like to see this paper as part of a broader project, I don't think it is strong enough and analytical enough to stand as a literature review project in and of itself.

Author Response

REVIEWER 1 COMMENTS

RESPONSE

Ensure that all references are relevant to the contents of the manuscript

All references have been cross-checked to establish conformity

Basic needs and strategies of refugees must be considered as an integral part of survival and therapy.

This has been captured and supported with scholarly information

Trauma recovery pathways and refugees were not defined in the study

These keywords have been defined and explained for better comprehension of the subject matter.

There should be an emphasis on therapy and supportive care being trauma informed

This has also been highlighted in the study

I was very excited to see the topic for this article and I would like to command the authors for undertaking this research and literature review, as it is important to assess trauma care responses for refugees and immigrants. However, this article did not go as deep as it should have done, even as a literature review article. 

There were only 9 academic articles that made it through the elimination, some of which was avoidable (52 articles out of 61 articles eliminated because full-text was not available or because they did not refer to "trauma recovery frameworks?") and then those 9 articles and their findings were not discussed well or analyzed, but simply grouped under very general categories.

This piece almost reads like an exercise in how to write a short literature review, without going into deeply the actual topic of trauma care for immigrants and refugees to pull important findings.

Thank you very much for this deep and thought-provoking comments. Authors have carefully considered and addressed (incorporated all suggestion) in line with the objective of the preliminary scoping review.

On the elimination of articles because full text could not be reached, we provide explanation – specifically mention that authors of the papers were emailed three times without response.

We also introduced some analysis to the findings as suggested by you (see diagnosis and Therapy section). As described in the objective statement of the study, we sought to bring out the prevailing general pathways as a rapid preliminary review, this accounts for why deep analysis was limited. However, We have now incorporated deep analysis into sections pointed out.

Reviewer 2 Report

It is important to add to the analysis that Basic Needs Strategies (food, shelter, safety, family reunification, transportation, employment and migrant/refugee legal documentation) precede and are more urgent than trauma treatment and therapy. This is an oversight that needs to be addressed. Refugees and migrants who lack attention to their basic needs will be too preoccupied with survival for psychological trauma care, although early treatment is important.

In addition, therapy and supportive care must be "trauma-informed" and based on evidence-based models of care proven to be effective with survivors of other types of trauma such as sexual assault, combat, kidapping and physical or psycholgical abuse.

Author Response

Reviewer 2

Response

It is important to add to the analysis that Basic Needs Strategies (food, shelter, safety, family reunification, transportation, employment and migrant/refugee legal documentation) precede and are more urgent than trauma treatment and therapy. This is an oversight that needs to be addressed. Refugees and migrants who lack attention to their basic needs will be too preoccupied with survival for psychological trauma care, although early treatment is important.

In addition, therapy and supportive care must be "trauma-informed" and based on evidence-based models of care proven to be effective with survivors of other types of trauma such as sexual assault, combat, kidapping and physical or psycholgical abuse.

Thank you so much for pointing this out. We have included sentences to address this in the Diagnosis and therapy subsection under findings and discussion.

This perspective is very important and authors have incorporated it in the discussion under diagnosis and therapy.

Reviewer 3 Report

This manuscript addresses the literature on "trauma recovery pathways" among "refugees." Unfortunately, the paper is plagued with multiple problems. To begin, the above key terms (in quotation marks) are not defined. The authors never define "trauma recovery pathways", which they also refer to as trauma recovery frameworks and approaches, which are defined either. With regard to "refugees", the authors begin by providing statistics on internally displaced persons, which is a different population from refugees.

Further, the inclusion and exclusion criteria for study selection are not explicit. 

In Table 1, the methods of the studies are insufficiently described. 

Author Response

Reviewer 3

This manuscript addresses the literature on "trauma recovery pathways" among "refugees." Unfortunately, the paper is plagued with multiple problems. To begin, the above key terms (in quotation marks) are not defined. The authors never define "trauma recovery pathways", which they also refer to as trauma recovery frameworks and approaches, which are defined either. With regard to "refugees", the authors begin by providing statistics on internally displaced persons, which is a different population from refugees.

Further, the inclusion and exclusion criteria for study selection are not explicit. 

In Table 1, the methods of the studies are insufficiently described

The keywords "trauma recovery pathways" among "refugees." have been defined and explained for better comprehension of the subject matter.

This has been worked on under the methods section.

This has be rectified in the table under methods used

Round 2

Reviewer 1 Report

I will defer to the second reader on this manuscript. I still think that it is too limited and relies on a "preliminary scoping" review of just a few articles but at least the revisions helped give it some depth.  I think the authors need to check their definition of refugee, though, because they are using the Cartegena definition, which is not globally accepted and for whatever reason, not using the 1951 Refugee Convention definition that is globally applied.  The categories in that Convention refugee definition is what the US Refugee Act definition (therefore all refugees recognized by the US is dependent on), so if the goal is to study the US, you might as well use the definition that is globally accepted and locally used in legal terms.

Author Response

Thank you for your comments. We have updated our definition of refugees, using the 1951 refugee convention definition in the introduction.

In line with the study objective, and the recommendation that has been pointed by some reviewer comments, the included studies are the only ones that we think fit into the study goal. We have read a couple of other articles very carefully and do not think those articles fit into this review.

Reviewer 2 Report

The revisions have addressed the concerns raised.

Author Response

We really appreciate your insight in strengthening the paper

Reviewer 3 Report

The authors have satisfactorily addressed most of my previous comments. However, they still need to provide more details about the study methodologies in Table 1. Naming a research design or analysis method is insufficient. They need to describe what each study did in terms of research questions/hypotheses, sample, setting, data collection method, etc.

Author Response

We are happy to have been able to address your comments satisfactorily. We have included a description of the methodologies to give context to the table as suggested on pages 4 and 5